# Anthropometric Measurements, Sociodemographics, and Lifestyle Behaviors among Saudi Adolescents Living in Riyadh Relative to Sex and Activity Energy Expenditure: Findings from the Arab Teens Lifestyle Study 2 (ATLS-2)

**DOI:** 10.3390/nu14010110

**Published:** 2021-12-27

**Authors:** Hazzaa M. Al-Hazzaa, Shaima A. Alothman, Abdullah F. Alghannam, Alaa A. Almasud

**Affiliations:** Lifestyle and Health Research Center, Health Sciences Research Center, Princess Nourah Bint Abdulrahman University, Riyadh P.O. Box 47330, Saudi Arabia; shaima.alothman.pt@gmail.com (S.A.A.); AFAlghannam@pnu.edu.sa (A.F.A.); AAAlmasud@pnu.edu.sa (A.A.A.)

**Keywords:** adolescents, dietary habits, health behaviors, lifestyle behaviors, obesity, physical activity, sedentary behaviors, sleep

## Abstract

The aim of the study was to examine the anthropometric measurements, sociodemographics, and lifestyle behaviors among Saudi adolescents relative to sex and physical activity (PA). A random cross-sectional survey conducted on Saudi adolescents from secondary schools in Riyadh, using a multistage stratified cluster sampling technique. Measurements included demographics, weight, height, waist circumference, PA, sedentary behaviors (SB), sleep duration, and dietary habits using a validated questionnaire. A total of 1262 adolescents (16.4 ± 0.95 years; 52.4% males) were studied. Overweight/obesity was more than 40%. Physical inactivity among adolescents was 53%, which indicates some improvement over the past years, especially among females. More than 80% of adolescents had over three hours/day of screen time, with no significant sex differences. Insufficient sleep was highly prevalent with gender differences. A large proportion of the participants did not consume daily breakfast (65.7%), vegetables (73.2%), fruits (84.2%), or milk/dairy products (62.4%), whereas significant proportions of the adolescents consumed sugar-sweetened drinks, fast food, French fries/potato chips, cake/donuts, and chocolates/candy on at least three days or more per week. It was concluded that non-daily intake of breakfast and vegetables was significantly associated with lower PA. The updated information can aid in effectively planning and implementing promotional programs toward improving the lifestyle behaviors of Saudi adolescent.

## 1. Introduction

Adolescence represents a critical developmental period during which personal lifestyle behaviors are established [1]. It is a period that can significantly impact later health and disease [2]. Monitoring lifestyle behaviors, including physical activity and sedentary time, during adolescence is important. A recent study that tracked lifestyle habits from youth to adulthood found a link between persistent physical inactivity and higher risk of impaired glucose metabolism in adulthood [3]. With rising levels of inactivity, sedentary behaviors, and Western diet in Saudi Arabia [4,5], continuous monitoring of the youth’s lifestyle behaviors is essential, as physical activity (PA) and dietary habits play significant roles in health enhancement and disease prevention [6]. The World Health Organization’s (WHO) through the *Global Action Plan on Physical Activity 2018–2030* emphasizes the importance of regular national surveillance of the population’s PA [7]. In addition, Saudi Arabia’s *health Strategy*, in collaboration with the WHO, identified behavioral risk factors for non-communicable diseases (NCDs) as one of the key strategic approaches to be undertaken by the country [8]. Furthermore, the Saudi Ministry of Health [9], and 2030 Saudi Vision [10] both highlight the importance of adopting healthy lifestyle for maintaining health and enhancing prosperity of Saudi society.

The benefits of PA for the health and well-being of children and adolescents are now widely recognized [11]. Therefore, global public health authorities have published PA guidelines for children and adolescents that typically call for adolescents to achieve a minimum of 60 min per day of moderate- to vigorous-intensity PA [11]. In spite of this, daily physical activity was shown to decrease during adolescence and such a decrease is more marked among females [12]. A recent report indicated that more than 80% of school-aged adolescents worldwide do not achieve the minimum level of recommended daily PA [13]. In addition, according to the WHO, more than three-quarters of the world’s population is not meeting the global recommendations for health-enhancing PA [11], with a sex difference above 8% in favor of men [14]. Additionally, findings from recent studies conducted in different countries, including those from the Global School-based Health Survey (GSHS), showed that sex difference in PA levels also exists among adolescents [13,15,16,17].

Earlier research from the Arab Teens Lifestyle Study (ATLS) conducted 10 years ago on Saudi adolescents showed that over 45% of males and 78% of females were inactive [5]. Furthermore, 84% of males and over 91% of females had more than two hours per day of screen time [5]. Such findings clearly indicated that Saudi females seemed consistently at a higher risk of both physical inactivity and too much sitting time than males [4,5]. Additionally, previous findings related to Saudi adolescents displayed contrasting results relative to sex and school type. Saudi males in public schools were more active than those in private schools, whereas the opposite was true for Saudi females [5,18,19]. Additionally, a number of lifestyle behaviors demonstrated significant sex effects among Saudi adolescents, with some selected parameters having multiple interaction effects. Total activity energy expenditure and the sum of all vigorous-intensity PA showed significant interaction effects between gender and obesity status, whereas the sum of moderate-intensity activity energy expenditure exhibited significant interaction effects between sex and screen time [20]. Such findings in the past, which indicated lower levels of PA among females, may have changed in the present time, as public schools have recently begun to introduce structured physical education curricula for girls, and women now have more opportunities than ever to engage in PA, including exercising in many newly opened gyms for females only [21].

An interest has recently emerged in investigating sex differences in activity behaviors at younger ages, with the intent of striving for a sex-equitable society [15]. In Saudi Arabia, there has been a recent positive sociopolitical transformation. This has led to more autonomy and opportunities for Saudi females to engage in various aspects of societal life. The country instituted structured physical education curricula and programs for girls in schools in 2017. It also allowed women to drive and travel more independently in 2018, and granted licenses to open vast private fitness centers for women in the country. All these changes have allowed for more opportunities for women to be able to participate more in sport and physical activities. Therefore, there is a need for updated information on the levels of physical activity and sedentary behaviors among Saudi adolescents relative to sex status, in order to tailor future lifestyle-promotion programs for youths. The present paper aims to present findings on anthropometric measurements, sociodemographics, and lifestyle behaviors among Saudi adolescents living in Riyadh relative to sex and activity energy expenditure.

## 2. Methods

The present research was conducted during the period from mid October 2019 until early March of 2020. We avoided weeks where periodical school’s exams were given to students, as students behaviors may change during such periods. Detailed methods of this study have been published elsewhere [22]. However, the following provides a brief description of the study’s methods and procedures.

### 2.1. Ethical Approval

The Institutional Review Board (IRB) at the Princess Nourah Bint Abdulrahman University granted the ethical approval for conducting the present research (IRB log number 18-0075). Informed consent was secured from all participants as well as from the parents of those adolescents under the age of 18 years. Additional approvals were also obtained from the Ministry of Education (MOE) and from the officials of the selected schools. The research procedures were conducted according to the principles expressed in the Declaration of Helsinki.

### 2.2. Participants and Selection Procedures

To select our sample from public and private secondary schools in Riyadh city, we used a multistage stratified cluster random sampling technique [22]. The sample size was calculated assuming a population proportion that would yield the maximum possible sample size required with a proportion of 0.50, a confidence level of 95%, and an error margin of 5%. After additional 20% to the original calculated sample, the whole sample size for each gender group was computed to be 461 students, or 922 adolescents in total (males and females). The chosen sample was in proportion to the total number of students attending public and private schools. The sample was chosen randomly from the pools of secondary schools for boys and girls based on four geographical zones and the central part of Riyadh. Within each school, classes were randomly selected from each of the three grades (10, 11, and 12). Within the selected classes, all healthy students without any medical condition that preclude them from physical activity were invited to participate in the study. Figure 1 illustrate the selection process.

### 2.3. Anthropometric Measurements, Demographics, and Socioeconomic Status

Measurement of body weight was made to the nearest 100 g with minimal clothing and without shoes using calibrated medical scales (Seca medical scale, model 770, Seca, Hamburg, Germany). Height was assessed to the nearest centimeter using a calibrated measuring rod while the subject was in a full standing position without shoes. Body mass index (BMI) was computed as the ratio of weight in kilograms to height in meters squared. To classify overweight or obesity status among adolescents between the ages of 14 and 17 years, the International Obesity Task Force (IOTF) age- and sex-specific BMI cutoff reference standards were used [23]. For participants aged 18 years and older, the adult cutoff points of 25–29.9 kg/m^2^ was used to express overweight and 30 kg/m^2^ and higher to define obesity. In addition, waist circumference (WC) was measured to the nearest 0.1 cm using a non-stretchable measuring tape. Measurement was taken horizontally at navel level and at the end of a gentle expiration. Waist-to-height ratio (W/Ht-R) was calculated as the ratio between WC in centimeters and height in centimeters. A W/Ht-R cutoff point of 0.50 was used to define abdominal obesity in both males and females [24,25]. Body shape index was also calculated as WC/((BMI **^2/3^**) × (height **^1/2^**)). In addition, we assessed demographics and socioeconomic status (SES) including age, sex, maternal and paternal education, and family income in Saudi Riyal (SAR). USD 1 equals SAR 3.75.

### 2.4. Assessment of Lifestyle Behaviors

The Arab Teen Lifestyle Study (ATLS) questionnaire was used to collect a number of lifestyle variables, including PA/inactivity, sedentary behaviors, sleep duration, and dietary habits [5,26]. The validity and reliability of the questionnaire in assessing PA and other lifestyle habits were previously established among youth from 14–25 years of age [5,26,27,28].

#### 2.4.1. Physical Activity

The PA questionnaire gathers data on the frequency, duration, and intensity of light-, moderate- and vigorous-intensity PA during a typical week. The questionnaire covers several domains of activity including transport, fitness, household, and sporting and leisure-time PA. The ATLS instrument permits the calculation of the whole time spent in minutes on all activities per week, as well as time spent in moderate or vigorous-intensity PA. The questionnaire translates physical activity time into metabolic equivalent (METs) in minutes per week (METs-min/week) using the youths’ compendium of physical activities [29]. To determine the participants’ levels of PA, we used the total activity energy expenditure in METs-min/week and the METs-min/week spent in each of the moderate- and vigorous-intensity PA. Two calculated scores expressing the percentage of adolescents who met the daily PA recommendations were used. The first score is corresponding to one hour per day of moderate-intensity (four METs) activity. The second one is equivalent to one hour per day of moderate- to vigorous-intensity (six METs) PA. These two amounts of physical activity were then converted into METs-min per week, corresponding to 1680 METs-min/week (60 min per day × 7 days per week multiplied by 4 METs) and 2520 METs-min per week (60 min per day × 7 days per week multiplied by 6 METs), respectively.

#### 2.4.2. Sedentary Behaviors and Sleep Duration

The ATLS questionnaire also includes items related to time spent in sedentary activities. They collect information from the participants related to typical daily time spent on all kinds of screen, including time spent watching TV, playing video games, and computer and internet leisure use. Adolescents were asked to provide the average number of daily hours of screen time during both weekdays and weekends. To calculate the sedentary time cutoff hours, we used the maximum recommended sedentary time for youth of two hours per day [30]. However, due to the high percentage of screen time exhibited by adolescents, we also used a cutoff of below or above three hours of screen time.

In addition, sleep duration during both weekdays (school days) and weekends were assessed using the ATLS questionnaire. Adolescents were asked about the typical amount of sleep duration on weekdays and at weekends. Insufficient sleep (short sleepers) was defined as sleeping less than eight hours per night, according to the designation of the National Sleep Foundation for school-age adolescents aged 14–17 years [31].

#### 2.4.3. Dietary Habits

Ten specific questions related to the frequency of certain dietary habits during a typical (usual) week were included in the ATLS questionnaires. The questions asked the participants about how many times per week they have breakfast, vegetables (cooked and uncooked), fruit, milk/dairy products, sugar-sweetened drinks (including soft drinks), fast food, cake/donuts, and chocolates/candy. The items covered healthy and unhealthy dietary habits. The adolescents can chose an answer ranging from zero intake (never) to a maximum intake of seven days per week (every day). The questions related to dietary habits that were included in ATLS questionnaire were sown to be valid and reliable in an earlier study [32].

### 2.5. Data and Statistical Analyses

Data were entered into an SPSS data file, and then checked, and analyzed using SPSS Statistics 22 (IBM, Chicago, IL, USA). To avoid over-reporting, PA scores and sedentary time were cleaned and truncated at realistic levels [5,26]. The maximum total time spent on PA per week was truncated at 28 h per week or four hours of maximum PA per day [5,26]. The maximum number of stair climbing undertaken by students per day was capped at 30 floors. The joint time of TV viewing, computer use, and internet time was truncated at 16 h per day [5,26]. A *t*-test for independent samples or chi-squared test for the proportion were used to test the differences between males and females in selected continuous or categorical variables, respectively. We also used multivariable analysis (MANCOVA) with Wilks’ Lambda tests for selected anthropometric and lifestyle variables stratified by gender and activity levels while controlling for age. In addition, logistic regression analysis of selected lifestyle behaviors and overweight/obesity status relative to activity energy expenditure in METs-min/week was used, while adjusted for age, sex, and sociodemographic factors. The level of significance was set at a value of ≤0.05.

## 3. Results

We collected data from 1262 participants (52.4% male). The response rate was extremely high, reaching 99.2%. Table 1 presents the anthropometric measurements and the sociodemographic characteristics of the participants relative to sex. The mean age for the participating adolescents was 16.4 ± 0.95 years (14–19 years). As expected, there were significant differences (*p* < 0.001) between males and females in body weight, height, BMI, WC, W/Ht-R, and body shape index. Based on the IOTF cutoff values, more than 40% of adolescents were either overweight or obese, and the prevalence of overweight or obesity in males (47.3%) was significantly (*p* < 0.001) higher than that of females (32.8%). Nearly 75% of adolescents came from public schools in Riyadh. There were significantly (*p* < 0.001) more males in public schools and more females in private schools. About 41% of fathers and 39.1% of mothers had university degrees, with a significant (*p* = 0.038) trend for higher levels of fathers’ (but not mothers’) education among students in private schools. Over 37% of the students’ families earned more than SAR 20,000 (USD 5326) per month, with significantly higher family income for those adolescents attending private schools.

Table 2 shows the proportions (%) of Saudi adolescents who exceeded certain cutoff values for overweight/obesity and selected lifestyle behaviors. Overall, more than 40% of adolescents were either overweight or obese. More than 80% of adolescents had over three hours per day of screen time, with no significant (*p* = 0.363) differences between males and females. Nearly 70% of the participants were not getting eight hours of sleep per night, with males having more insufficient sleep duration than females (*p* < 0.001). As to the levels of health-enhancing PA, 53.4% of the sample did not obtain the recommended activity energy expenditure of 1680 METs-min/week (which is equivalent to one hour of daily moderate-intensity PA). Approximately 66% of the sample did not have breakfast every day before going to school. Moreover, a large percentage of adolescents did not consume daily vegetables, fruit, or milk/dairy products. On the other hand, about 58% of the adolescents were consuming sugar-sweetened drinks, 48.6% eating fast food, 44.5% consuming French fries/potato chips, 40.6% having cake/donuts, and 56% consumed chocolates/candy on at least three days or more per week.

The results of activity energy expenditure (in METs-minutes/week) expended in different types of PA by Saudi adolescents are shown in Table 3. There were no significant differences between males and females in activity energy expenditure (in METs-min/week) spent walking, climbing the stairs, or doing martial arts or resistance training, while males spent significantly more time on activities such as jogging, cycling, swimming, moderate-intensity sports, and vigorous-intensity sports. On the other hand, females spent significantly more time on activities like household chores and dancing compared with males. In addition, males were significantly (<0.001) more active in their total vigorous-intensity PA, whereas females showed significantly (<0.001) higher levels in their total moderate-intensity PA. However, there was no difference in total PA relative to sex. Furthermore, overall, inactivity prevalence (<1680 METS-min/week) was 53.4%, with more males (58%) than females (48.5%) being inactive.

Other findings that are not shown in the tables include that more females (76.6%) than males (31.4%) were doing PA at home, with females (64.9%) being more likely than males (35.8%) to exercise alone. Males were active mostly for health (47.8%) or weight loss (23.0%), while the respective proportions in females exercising for health or weight loss were 52.2% and 26.1%. As to reasons for being inactive, these were lack of time (50% for males and 52.2% for females) and lack of suitable place (15.1% for males and 11.5% for females).

Table 4 presents the proportions of sociodemographic and lifestyle factors relative to activity energy expenditure levels among the Saudi adolescents. There was no significant difference between public and private schools in activity levels. Additionally, there were no significant differences between active and inactive adolescents relative to sociodemographic factors, overweight/obesity status, screen time, or sleep duration. However, 40% of the active adolescents consumed breakfast daily, whereas only 29.3% of inactive adolescents consumed breakfast daily (*p* < 0.001). Active participants consumed more daily vegetables (33.2% versus 21.5%, *p* < 0.001) and daily fruit (19.8% versus 12.7%, *p* = 0.001) than inactive participants. There were no significant differences between low and high active adolescents in less healthy dietary intakes such as sugar-sweetened drinks, fast food, French fries/potato chips, cake/donuts, or chocolates/candy.

Table 5 exhibits the findings of multivariable analysis of anthropometric and lifestyle variables stratified by sex and activity levels while controlling for age. There were no significant differences relative to activity levels in anthropometric measurement, screen time, or sleep duration. However, while for controlling age, active adolescents showed significantly higher intakes of breakfast (*p* < 0.001), vegetables (*p* < 0.001), fruit (*p* < 0.001), and milk/dairy products (*p* = 0.019).

Table 6 displays the results of logistic regression analysis of selected lifestyle behaviors and overweight/obesity status relative to activity energy expenditure (METs-min/week) among Saudi adolescents, while adjusted for age, sex, and sociodemographic factors. Activity energy expenditure cutoff scores (active versus inactive) showed that non-daily intake of breakfast (*p* = 0.001) and vegetables (*p* < 0.001) were significantly associated with lower PA levels (aOR and 95% CI: 0.615 (0.463–0.817), and 0.555 (0.403–0.764) for breakfast and vegetable intake, respectively.

## 4. Discussion

The aim of the present study was to assess the anthropometric measurements, sociodemographics, and lifestyle behaviors among Saudi adolescents attending high schools in Riyadh relative to sex and activity energy expenditure. The findings indicated that more than 40% of adolescents were either overweight or obese, and the prevalence of overweight or obesity in males was significantly higher than that of females. The prevalence of physical inactivity among Saudi adolescents was 53%, which shows some improvement compared with previous studies [5,18]. Such improvement in the levels of PA is seen more in females than in males. Females spent significantly more time than males in moderate-intensity activities like household chores and dancing compared with males, while male adolescents spent significantly more time than females in vigorous-intensity physical activity. More than 80% of adolescents had over three hours per day of screen time, with no significant differences relative to sex. Additionally, nearly 70% of the participating adolescents were not getting the recommended eight hours of sleep per night, with males having more insufficient sleep duration than females. A large proportion of the participants did not consume daily breakfast, vegetables, fruit, or milk/dairy products. On the other hand, significant proportions of the adolescents consumed sugar-sweetened drinks, fast food, French fries/potato chips, cake/donuts, and chocolates/candy on at least three days or more per week. Finally, the results of logistic regression while adjusted for age, sex, and sociodemographic factors showed that non-daily intake of breakfast and vegetables were significantly associated with lower physical activity levels.

The finding that more than 53% of Saudi adolescents are considered inactive is similar to those findings reported for adolescents in many countries. Data from 298 school-based surveys in 146 countries, territories, and areas including 1.6 million students aged 11–17 years conducted in 2016 indicated that the majority of adolescents do not meet current PA guidelines [13,14]. The prevalence of insufficient PA significantly decreased between 2001 and 2016 for boys, whereas no significant change was seen for girls [13,14]. Recent guidelines from the WHO, though, recommend that children and adolescents aged 6–17 years engage in at least 60 min of moderate- to vigorous-intensity physical activity every day [11]. A recent study conducted on Saudi adolescents aged 13–14 years in the city of Arar (in the northern part of the country) found a high prevalence of physical inactivity (92.7%) among the participants [33]. However, this study used a different PA questionnaire (the PA Questionnaire for Older Children) than the one used in our current study.

The present study showed that a large proportion of females practiced dancing as a form of PA. Local dancing is popular among young Saudi females. Data from the National Health and Nutrition Examination Survey 1999–2006 in the US indicated that the attributable proportion was higher among men than women for sports and higher among women than men for walking and dancing [34]. A dance-based intervention study aimed to increase the objectively assessed weekday minutes of moderate- to vigorous-intensity PA among adolescent girls in Bristol, UK, showed that the intervention was enjoyed by the girls [35].

The present study found that Saudi girls are not significantly different than males in total activity energy expenditure. This finding was somewhat surprising and contrary to earlier findings on young Saudi females [4,5,18]. Additionally, numerous studies around the world have shown that adolescent females were less active than males. In a large study conducted on children and adolescents aged 5–17 years in Latin America, sex differences in PA (≥60 min/day) and sedentary behaviors (≥3 h/day) were evident [15]. The largest sex difference in PA between boys and girls was reported in Uruguay (13.5%) and the lowest in Jamaica with 1.8% [15].

Among Japanese children and adolescents, boys were more physically active and took more steps/day than girls. They were significantly more likely to meet physical activity guidelines than girls [36]. Additionally, a cross-sectional study that included high school students in Florence, Italy, indicated that the prevalence of participants who reached the recommended levels was lower among girls compared to boys. According to this study, the number of perceived barriers to PA was higher among girls than among boys [17]. In addition, data from Quebec Adiposity and Lifestyle Investigation in Canadian Youth showed that girls were more likely than boys to belong to the trajectory with lower moderate to vigorous intensity PA means (OR: 6.45; 95%; CI: 3.08 to 13.49), and at the same time were less likely to belong to the trajectory with higher screen time (OR: 0.47; 95%; CI: 0.23 to 0.97) [37]. The International Children’s Accelerometry Database showed that boys were less sedentary and more active than girls at all ages from 3 to 18 years [38].

Research elsewhere showed that male adolescents are likely to be motivated to be physically active as a result of the competitive aspects of the activity, but girls are attracted to PA by the social opportunities provided by the sport [39]. In addition, the motivating factors for PA among males appear to be more intrinsic, such as improving health and well-being, preventing disease or risks, enhancing body shape, and being competitive; however, a combination of extrinsic and intrinsic factors such as emotional support, social aspects, sense of well-being, and positive body image seem to motivate females to be more active [40]. However, previous research among adults revealed that sex differences in overall PA were not obvious, but men were shown to exercise more vigorously [41]. The present study showed that adolescent males exercised more vigorously than females.

The increased levels of PA among Saudi females compared to previous local studies [4,5,18] could be the consequence of state policies that were introduced recently to promote more active and independent Saudi females. Quality of life and enhancing women’s health and well-being were among the objectives of the policies that were introduced by the Ministry of Health [9], and Saudi Vision 2030 [10]. In just a few years since the policies were pronounced, a huge number of fitness centers for females have been opened. Additionally, physical education curricula were instituted for all-girls’ schools; before this time, physical education (PE) was absent in public schools and optional in private schools for girls. In fact, a recent study conducted on adult females attending fitness centers in Saudi Arabia showed that the total activity energy expenditure spent by those motivated females was very high and equaled 3819.5 METs-min/week [21].

It is also possible that Saudi females are more motivated to engage in PA for weight loss and enhancing health. A recent cross-sectional study that was conducted among university students in Saudi Arabia showed that nearly 60% of female students were using health-related applications compared to 49% of male students, and that the most frequent aim of using mobile health applications were tracking PA (72.5%) followed by counting calorie intake wit 44% [42].

Public health interventions, when successfully implemented, are likely to improve PA, fitness, and health as well as reduce the prevalence of overweight and obesity in youth [43]. State policy through legislation, such as those initiated in Saudi Arabia, plays an essential role in promoting PA among school students, as mandating PE classes and frequent recess can increase students’ opportunities to engage in regular PA at school [44]. Among American middle and high school students, it was shown that state PE policies tended to be more closely related to girls’ weekly PE attendance and levels of physical activity than boys’ weekly PE attendance and PA status. It is possible that girls are less likely to take PE as an elective course; therefore, mandating PE increased girls’ PE time more substantially than boys [45]. Additionally, data from the US National Youth Risk Behavior Survey indicate that state PE policies impacted high school students’ PE class attendance, with larger effects on female students [46].

The present study revealed that there was no significant difference between active and inactive adolescents relative to sociodemographic factors. Socioeconomic status (SES) has been identified as an important correlate of PA and a healthy lifestyle [47]. Results of a recent study involving German children and adolescents found that participants with a higher parental SES were more physically active than participants with lower SES [48]. The differences regarding parental SES, though, were much more apparent for organized sports than for unorganized sports [48]. Organized sports in youth are usually dependent on facility availability, as it was previously reported that communities with higher SES were more likely to have more than one PA facility [49]. Contrary to previous studies, findings from global school-based surveys in 146 countries and territories did not show that the prevalence of insufficient PA activity in school-going adolescents increased with country income [13].

Adolescent obesity is perhaps the most serious public health challenge of recent years [50], and its prevalence among Saudi children and adolescents has been rising over the last few decades [51]. The latest national survey shows that the prevalence of being overweight or obese among Saudi youth aged 14–24 years was 54.1% and 51.6% for males and females, respectively [52]. The current study showed no significant differences among participants relative to activity levels in anthropometric measurement. However, the relationship between body weight and PA is influenced by numerous confounding variables, such as sedentary behaviors and dietary habits. Contrasting findings have been reported regarding the associations between PA and adolescents’ obesity. A number of studies have shown an inverse association between PA and weight status in adolescents [53,54,55]. A prospective study observed that higher levels of moderate- to vigorous-intensity PA were associated with lower levels of fat mass in early adolescence [55].

On the other hand, self-reported PA scores were not significantly different among obese and non-obese children [56]. In addition, among Iranian adolescents, no significant association between PA and body weight status was detected [57]. It seems that different obesity levels may influence the associations between body weight and PA. In one study, subgroups of children with overweight, obesity, and morbid obesity were studied, and the findings revealed that children with obesity perform less PA and more sedentary behaviors compared to children with morbid obesity [58], as they probably were more likely trying to lose weight through increasing their PA.

More than 80% of adolescents had over three hours per day of screen time, with no significant differences between males and females. Additionally, nearly 70% of the participating adolescents were not getting eight hours of sleep per night, with males having more insufficient sleep duration than females. An earlier study conducted on Saudi adolescents revealed that very high proportions of males (84%) and females (91.2%) were sedentary, defined as having more than two hours per day of screen time [5]. Adolescents from a northern city in Saudi Arabia (Arar) were found to have nearly six hours of daily sedentary time during weekdays and 7.8 h during weekends [33]. Furthermore, a large study of US adolescents showed that the prevalence of sitting while viewing screens for at least two hours per day was high and the estimated sitting time totaled 8.2 h per day among adolescents [59]. With increased reliance on computers and the internet, young people are prone to substitute PA for sedentary time. A recent Norwegian research tracked PA from 2005/2006 to 2011/2012 using accelerometers and found that both children and adolescents replaced time spent in light PA for time spent being sedentary [60].

Clear guidelines on sedentary sitting time for children do exist. The Canadian 24-Hour Movement Guidelines for Children and Youth recommends that children and youth should not have more than two hours of recreational screen time daily [30]. However, the WHO recommends reducing sedentary behaviors across all age groups and abilities (11). The concept of sedentary behaviors is now recognized as a different entity from physical inactivity [61], and sedentary behaviors are known to associate independently with metabolic risks in children [62]. In fact, physical inactivity and sedentary behaviors are believed to be associated with different adverse health outcomes [63]. Excessive screen viewing time among adolescents also appears to be related to unfavorable cardiovascular disease risks [64]. Sedentary behaviors are also an important risk factor for increased adiposity among children and adolescents aged 7–15 years [65]. Higher levels of screen time and lower levels of PA were related to lower life satisfaction and higher psychosomatic complaints among adolescents from high-income countries [66]. Moreover, unhealthy behaviors appear to aggregate in this group of Saudi adolescents, as healthful dietary habits (breakfast, fruit, vegetables, and milk/dairy) were associated mostly with PA, whereas unhealthful dietary habits (higher consumption of sugar-sweetened drinks, fast food, cake/donuts, and energy drinks) were related most to screen time [67].

The present research did not find any association between levels of PA and sleep duration among Saudi adolescents. An earlier study conducted on Saudi adolescents showed that PA levels were significantly associated with daily sleep of eight hours or more [68]. However, there was no difference between the amount of sleep at night in the findings of the present study and those reported in previous research [69]. Previous studies have shown that active adolescents were more likely to have sufficient sleep duration compared with inactive peers [70,71,72]. However, other studies reported no association [73,74] or an inverse relationship [75,76] between PA and sleep duration. Exercise is believed to promote sleep through different hypotheses, including energy conservation, tissue restoration, and temperature down-grading [77].

### Strength and Limitations

This study has both strengths and limitations that should be acknowledged. Strengths include the fact that the sample size was adequate and representative of Saudi adolescents attending public and private secondary schools in Riyadh. In addition, the PA questionnaire covers all domains of PA, has been shown to be valid and reliable, and has been frequently used over the years in numerous previous studies. Additionally, there was a high response rate among Saudi adolescents in this study, and the study’s analysis adjusted for important confounders, such as age, gender, and sociodemographic factors.

The present study also has some notable limitations. The cross-sectional design precludes a causal inference for the relationship between physical activity and the selected variables in this study. We also recognize the limitations of self-reported lifestyle behaviors, which may have resulted in overestimating or underestimating of true activity energy expenditure, sedentary time, sleep duration, or dietary habits. Finally, due to recent state legislatives endowing women to be more active, females may have been overly motivated to engage in exercise at the time the study was conducted.

## 5. Conclusions

The findings indicated that more than 40% of adolescents were either overweight or obese, and the prevalence of overweight or obesity in males was significantly higher than that of females. The prevalence of physical inactivity was 53%, which shows some improvement compared with previous studies. Improvement in levels of PA is more apparent in females than in males. Females spent significantly more time than males in moderate-intensity activities like household chores and dancing, while male adolescents spent significantly more time than females in vigorous-intensity PA. More than 80% of adolescents had over three hours per day of screen time, with no significant differences relative to gender. Additionally, nearly 70% of the participating adolescents were not getting the recommended eight hours of sleep per night, with males having more insufficient sleep duration than females. A large proportion of the participants did not consume daily breakfast, vegetables, fruit, or milk/dairy products. On the other hand, significant percentages of the adolescents consumed sugar-sweetened drinks, fast food, French fries/potato chips, cake/donuts, and chocolates/candy on at least three days or more per week. Finally, the results of the logistic regression, adjusted for age, sex, and sociodemographic factors, showed that non-daily intake of breakfast and vegetables were significantly associated with lower PA levels. The findings of this study provide updated and important information for effectively planning and implementing promotional programs toward improving the lifestyles and well-being of Saudi adolescents.

## Figures and Tables

**Figure 1 nutrients-14-00110-f001:**
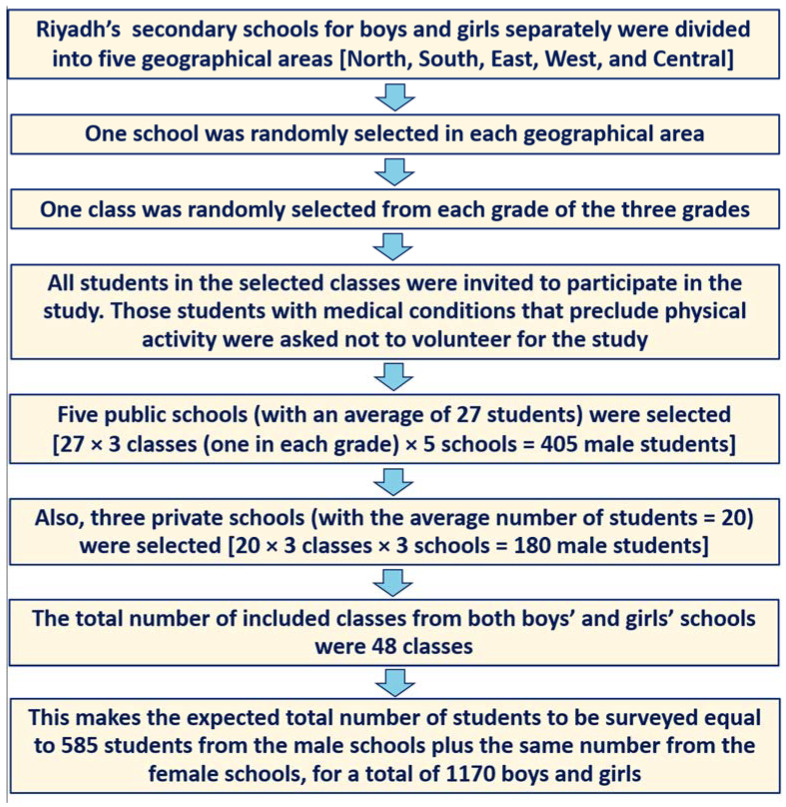
Flowchart showing the process of the students’ selection.

**Table 1 nutrients-14-00110-t001:** Anthropometric characteristics of the participants relative to sex.

Variable	All *n* = 1261	Male *n* = 660	Female *n* = 601	*p*-Value *
Age (year)	16.4 ± 0.95	16.4 ± 0.94	16.3 ± 0.96	0.050
Body weight (kg)	65.6 ± 20.9	73.2 ± 23.1	57.3 ± 14.3	<0.001
Body height (cm)	163.6 ± 8.9	169.7 ± 6.8	157.0 ± 5.5	<0.001
Body mass index (kg/m^2^)	24.3 ± 6.6	25.3 ± 7.4	23.2 ± 5.4	<0.001
Waist circumference (cm)	77.5 ± 16.5	83.6 ± 17.5	70.7 ± 12.4	<0.001
Waist-to-height ratio (W/Ht-R (%))	0.47 ± 0.09	0.49 ± 0.09	0.45 ± 0.08	<0.001
Body shape index **	0.73 ± 0.07	0.75 ± 0.06	0.70 ± 0.06	<0.001
Overweight + obesity (%) ***	40.5	47.3	32.8	<0.001
School type (%)				0.001
Public	74.9	78.8	70.5	
Private	25.1	21.2	29.5	
Father’s education (%)				0.038
Intermediate or less (≤9 years)	14.3	16.7	11.6	
High school	28.2	28.9	27.4	
University degree	40.9	38.0	44.1	
Postgraduate degree	16.6	16.4	16.9	
Mother’s education (%)				0.584
Intermediate or less (≤9 years)	21.5	20.0	23.0	
High school	29.2	29.7	28.7	
University degree	39.1	39.4	38.7	
Postgraduate degree	10.2	10.9	9.6	
Family income (%) ****				<0.001
SAR 10,000 or less	17.4	20.2	13.9	
SAR 10,001–20,000	42.2	48.1	34.9	
SAR 20,001–30,000	16.4	16.4	24.3	
SAR 30,001+	20.5	15.3	26.9	

Data are means ± standard deviations or percentage. * *t*-test for independent samples or Chi Squares tests for the proportion for the differences between males and females in continuous or categorical variables, respectively. ** Body Shape Index = WC/((BMI **^2/3^**) × (Height **^1/2^**)) *** Based on IOTF cut-off values (reference number 22) **** USD = 3.75 Saudi Riyal (SAR).

**Table 2 nutrients-14-00110-t002:** The proportions (%) of Saudi adolescents who exceeded certain cut-off values for overweight/obesity and selected lifestyle behaviors.

Variable	Criterion *	Proportion (%)			*p*-Value **
		All	Male	Female	
Overweight or obesity	BMI > cut-offs	40.5	47.3	32.8	<0.001
Abdominal obesity (W/Ht-R)	>0.50	31.7	40.9	21.4	<0.001
Screen time	>2 h/day	89.7	89.0	90.5	0.363
	>3 h/day	80.6	79.0	82.4	0.128
Nocturnal sleep duration	<8 h/night	69.1	73.6	64.3	<0.001
PA in Metabolic equivalent	<1680 METs-min/week	53.4	58.0	48.5	0.001
	<2520 METs-min/week	66.8	69.8	63.4	0.004
Breakfast intake at home	Non-daily	65.7	66.6	64.7	0.492
Vegetable intake	Non-daily	73.2	75.2	71.0	0.097
Fruit intake	Non-daily	84.2	84.1	84.2	0.970
Milk/dairy products	Non-daily	62.4	61.7	63.1	0.624
Sugar sweetened drink intake	≥3 day/week	57.5	62.5	52.1	0.001
Fast food intake	≥3 day/week	48.6	52.0	44.4	0.013
French fries/potato chips intake	≥3 day/week	44.5	42.7	46.6	0.225
Cake/donuts intake	≥3 day/week	40.6	37.4	44.1	0.039
Chocolates/candy intake	≥3 day/week	56.0	50.4	62.2	<0.001

* Overweight or obesity cut-offs are based on IOTF cut-off values [22]; waist-to-height ratio cut-offs are based [23,24]; screen time cut-offs are based on [29]; sleep duration cut-offs are based on [30]; and physical activity cut-offs are below or above 1680 METs-min/week (an equivalent of 60 min of daily moderate intensity physical activity), and 2520 METs-min/week (an equivalent of one hour of daily moderate to vigorous physical activity). ** Tests for the differences in proportions between males and females.

**Table 3 nutrients-14-00110-t003:** Activity energy expenditure (METs-minutes/week) expended in different types of physical activity by Saudi adolescents.

Variable	All (*n* = 1189)	Male (*n* = 591)	Female (*n* = 598)	*p* Value *
Walking (METs-min/week)	350.9 ± 14.8	334.9 ± 20.3	366.6 ± 21.6	0.286
Stair Stepping (METs-min/week)	86.6 ± 0.93	85.3 ± 1.4	87.9 ± 1.3	0.145
Jogging (METs-min/week)	363.0 ± 20.9	409.7 ± 32.3	317.1 ± 26.8	0.027
Cycling (METs-min/week)	64.0 ± 7.0	89.7 ± 12.9	38.7 ± 5.6	<0.001
Swimming (METs-min/week)	119.4 ± 11.8	149.5 ± 20.1	89.7 ± 12.6	0.012
Martial art (METs-min/week)	39.5 ± 7.2	25.7 ± 9.6	53.1 ± 10.7	0.057
Resistance training (METs-min/week)	101.3 ± 10.7	113.8 ± 17.5	88.9 ± 12.3	0.244
Household (METs-min/week)	275.4 ± 14.7	198.3 ± 16.4	351.2 ± 23.9	<0.001
Dancing (METs-min/week)	351.7 ± 21.8	15.2 ± 4.5	700.5 ± 39.0	<0.001
Moderate-intensity sports (METs-min/week)	155.2 ± 10.6	206.3 ± 18.2	104.9 ± 10.9	<0.001
Vigorous-intensity sports (METs-min/week)	498.0 ± 29.7	735.5 ± 51.6	264.6 ± 26.9	<0.001
Sum of all moderate-intensity physical activity (min/week)	298.1 ± 8.8	218.4 ± 9.5	376.6 ± 14.0	<0.001
Sum of all vigorous-intensity physical activity (min/week)	169.3 ± 6.6	213.7 ± 10.9	125.6 ± 7.1	<0.001
METs-min/week from moderate-intensity physical activity **	1134.1 ± 35.2	756.2 ± 33.4	1520.1 ± 58.0	<0.001
METs-min/week from vigorous-intensity physical activity ***	1271.8 ± 49.3	1609.2 ± 81.5	936.4 ± 52.5	<0.001
METs-min/week from total physical activity	2406.3 ± 67.0	2365.4 ± 99.3	2456.8 ± 90.3	0.463
Activity levels (%): Low (<1680 METS-min/week)	53.4	58.0	48.5	0.004
Moderate (1680–2519 METs-min/week)	13.4	11.8	14.9	
High (≥2520 METs-min/week)	33.3	30.2	36.6	

Data are means and standard errors. MET = metabolic equivalent. * *t*-test for independent samples for the differences between active and inactive females. ** Include activities that are less than 6 METs, such as walking, dance, household chores, and moderate-intensity sports like volleyball, table tennis. *** Include activities that are 6 + METs, such as Jogging, stair stepping, cycling, swimming, self-defense, resistance training, and vigorous sports like soccer, basketball, and handball, single tennis.

**Table 4 nutrients-14-00110-t004:** The proportions (%) of socio-demographic and lifestyle factors relative to activity energy expenditure levels among the participating adolescents.

Variable	Activity Energy Expenditure Levels		*p*-Value *
	Low Active (<1680 METs-min/week)	High Active (≥1680 METs-min/week)	
School type (%)			0.514
Public	53.9	46.1	
Private	51.8	48.2	
Father’s education (%)			0.071
Intermediate or less	15.1	14.1	
High school	26.5	30.3	
University degree	43.2	36.9	
Postgraduate degree	15.1	18.8	
Mother’s education (%)			0.510
Intermediate or less	20.9	23.0	
High school	30.8	27.7	
University degree	38.9	38.3	
Postgraduate degree	9.5	11.0	
Family income (%) **			0.834
SAR 10,000 or less	17.3	18.1	
SAR 10,001–20,000	42.9	40.4	
SAR 20,001–30,000	20.0	19.9	
SAR 30,001+	19.8	21.5	
Overweight or obesity (%)			0.354
Non-overweight/non-obesity	58.7	61.3	
Overweight/obesity	41.3	38.7	
Screen time (%)			0.239
≤3 h/day	18.5	21.2	
>3 h/day	80.5	78.8	
Sleep duration			0.681
<8 h/night	68.6	69.7	
≥8 h/night	31.4	30.3	
Breakfast intake (%)			<0.001
Non-daily intake	70.7	60.0	
Daily intake	29.3	40.0	
Vegetable intake (%)			<0.001
Non-daily intake	78.5	66.8	
Daily intake	21.5	33.2	
Fruit intake (%)			0.001
Non-daily intake	87.3	80.2	
Daily intake	12.7	19.8	
Milk/dairy products intake (%)			0.095
Non-daily intake	64.4	59.8	
Daily intake	35.6	40.2	
Sugar sweetened drink intake (%)			0.089
1–2 days/week	39.8	46.0	
3–4 days/week	22.3	19.2	
5+ days/week	37.9	34.8	
Fast food intake (%)			0.537
1–2 days/week	51.3	52.4	
3–4 days/week	27.5	28.8	
5+ days/week	21.2	18.8	
French fries/potato chips intake (%)			0.674
1–2 days/week	54.4	56.8	
3–4 days/week	24.6	22.9	
5+ days/week	21.0	20.3	
Cake/donuts intake (%)			0.414
1–2 days/week	60.6	57.7	
3–4 days/week	19.9	19.7	
5+ days/week	19.5	22.6	
Chocolates/candy intake (%)			0.086
1–2 days/week	46.4	40.9	
3–4 days/week	22.6	22.6	
5+ days/week	31.0	36.5	

* Chi-Square tests for the differences in proportions between low and high active. ** SAR = Saudi Riyal = USD 3.75.

**Table 5 nutrients-14-00110-t005:** Multivariable analysis for selected anthropometric and lifestyle variables stratified by sex and activity levels while controlling for age.

Variable	Sex	Activity Levels (*n* = 1108)		*p*-Value *
		Low Active	High Active	
Age (years)	Male	16.4 ± 0.94	16.3 ± 0.95	Activity levels: 0.493 Gender: 0.336 Activity levels by gender interaction: 0.995
	Female	16.4 ± 0.98	16.4 ± 0.91	
	All	16.4 ± 0.96	16.3 ± 0.93	
Body weight (kg)	Male	73.5 ± 23.2	71.3 ± 21.4	Activity levels: 0.988 Gender: <0.001 Activity levels by gender interaction: 0.052
	Female	56.0 ± 13.4	58.1 ± 14.8	
	All	65.4 ± 21.3	64.2 ± 19.3	
BMI (kg/m^2^)	Male	25.4 ± 7.5	24.7 ± 6.9	Activity levels: 0.868 Gender: <0.001 Activity levels by gender interaction: 0.077
	Female	22.7 ± 5.2	23.5 ± 5.5	
	All	24.1 ± 6.6	24.1 ± 6.2	
Waist circumference (cm)	Male	83.8 ± 18.3	82.7 ± 15.9	Activity levels: 0.647 Gender: <0.001 Activity levels by gender interaction: 0.466
	Female	70.5 ± 13.2	70.8 ± 12.0	
	All	77.6 ± 17.4	76.3 ± 15.1	
Body shape index	Male	0.7 ± 0.07	0.75 ± 0.05	Activity levels: 0.247 Gender: <0.001 Activity levels by gender interaction: 0.029
	Female	0.71 ± 0.7	0.69 ± 0.06	
	All	0.73 ± 0.07	0.72 ± 0.6	
Screen time (hours/day)	Male	5.2 ± 2.4	4.9 ± 2.6	Activity levels: 0.418 Gender: <0.001 Activity levels by gender interaction: 0.309
	Female	5.7 ± 2.7	5.7 ± 2.9	
	All	5.4 ± 2.5	5.3 ± 2.8	
Sleep duration (hours/night)	Male	7.1 ± 1.6	6.9 ± 1.5	Activity levels: 0.096 Gender: <0.001 Activity levels by gender interaction: 0.551
	Female	7.5 ± 1.7	7.4 ± 1.8	
	All	7.3 ± 1.7	7.1 ± 1.7	
Breakfast intake (day/week)	Male	3.58 ± 2.7	4.32 ± 2.7	Activity levels: <0.001 Gender: 0.300 Activity levels by gender interaction: 0.308
	Female	3.57 ± 2.9	3.97 ± 2.9	
	All	3.58 ± 2.8	4.13 ± 2.8	
Vegetable intake (day/week)	Male	3.57 ± 2.3	4.48 ± 2.3	Activity levels: <0.001 Gender: 0.386 Activity levels by gender interaction: 0.792
	Female	3.48 ± 2.5	4.30 ± 2.5	
	All	3.53 ± 2.4	4.38 ± 2.4	
Fruit intake (day/week)	Male	2.97 ± 2.3	3.64 ± 2.3	Activity levels: <0.001 Gender: 0.043 Activity levels by gender interaction: 0.484
	Female	2.60 ± 2.2	3.46 ± 2.4	
	All	2.80 ± 2.3	3.54 ± 2.3	
Milk/dairy products intake (day/week)	Male	4.37 ± 2.5	4.91 ± 2.3	Activity levels: 0.019 Gender: 0.001 Activity levels by gender interaction: 0.256
	Female	4.04 ± 2.6	4.24 ± 2.8	
	All	4.21 ± 2.6	4.55 ± 2.6	
Sugar sweetened drink intake (day/week)	Male	3.8 ± 2.5	3.53 ± 2.5	Activity levels: 0.173 Gender: 0.016 Activity levels by gender interaction: 0.666
	Female	3.37 ± 2.5	3.20 ± 2.6	
	All	3.60 ± 2.5	3.36 ± 2.5	
Fast food intake (day/week)	Male	2.88 ± 1.9	2.94 ± 1.9	Activity levels: 0.436 Gender: 0.275 Activity levels by gender interaction: 0.196
	Female	2.89 ± 1.9	2.63 ± 1.9	
	All	2.89 ± 1.9	2.77 ± 1.9	
French fries/potato chips intake (day/week)	Male	2.52 ± 1.9	2.46 ± 2.0	Activity levels: 0.344 Gender: 0.003 Activity levels by gender interaction: 0.636
	Female	2.94 ± 2.0	2.76 ± 2.1	
	All	2.72 ± 2.0	2.62 ± 2.1	
Cake/donuts intake (day/week)	Male	2.22 ± 2.0	2.30 ± 2.1	Activity levels: 0.187 Gender: <0.001 Activity levels by gender interaction: 0.506
	Female	2.71 ± 2.1	2.95 ± 2.2	
	All	2.45 ± 2.1	2.65 ± 2.2	
Chocolates/candy intake (day/week)	Male	2.8 ± 2.1	3.10 ± 2.2	Activity levels: 0.064 Gender: <0.001 Activity levels by gender interaction: 0.851
	Female	3.68 ± 2.4	3.91 ± 2.5	
	All	3.22 ± 2.3	3.53 ± 2.4	

Data are means and standard deviations. Activity levels were based on above or below activity energy expenditure of 1680 METs-min/week. * Wilks’ Lambda *p* values for multivariable tests: age < 0.001; gender < 0.001; activity levels < 0.001; and gender by activity levels interactions < 0.001.

**Table 6 nutrients-14-00110-t006:** Results of logistic regression analysis of selected lifestyle behaviors and overweight/obesity status relative to activity energy expenditure (METs-min/week) among Saudi children, while adjusted for age, sex, and socio-demographic factors.

Variable	High Versus Low Active *			
	aOR	(95% CI)	SEE	*p*-Value
Age (younger age = ref)	0.947	0.8–251.087	0.071	0.439
Sex (female = ref)	0.763	0.577–1.009	0.143	0.058
Father education (low education = ref)	1.022	0.865–1.208	0.085	0.797
Mother education (low education = ref)	0.966	0.818–1.141	0.085	0.686
Family income (low income = ref)	0.965	0.834–1.116	0.074	0.628
Overweight or obesity (Overweight/obesity = ref)	1.00			
Non-overweight/non-obesity	1.069	0.733–1.559	0.192	0.728
Waist-to-height ratio (<0.50 = ref)	1.00			
≥0.50	1.195	0.798–1.790	0.206	0.386
Screen time (low screen time = ref)	1.00			
High screen time	1.145	0.823–1.593	0.168	0.421
Sleep duration (insufficient sleep = ref)	1.00			
Sufficient sleep	1.141	0.856–1.520	0.146	0.368
Breakfast intake (daily intake = ref)	1.00			
Non-daily intake	0.615	0.463–0.817	0.145	0.001
Vegetable intake (daily intake = ref)	1.00			
Non-daily intake	0.555	0.403–0.764	0.164	<0.001
Fruit intake (daily intake = ref)	1.00			
Non-daily intake	0.784	0.536–1.149	0.195	0.212
Milk/dairy products intake (daily intake = ref)	1.00			
Non-daily intake	1.081	0.813–1.437	0.145	0.591
Sugar sweetened drink intake (daily intake = ref)	1.00			
Non-daily intake	1.110	0.795–1.550	0.170	0.539
Fast food intake (daily intake = ref)	1.00			
Non-daily intake	1.147	0.658–2.001	0.284	0.628
French fries/potato chips intake (daily intake = ref)	1.00			
Non-daily intake	1.356	0.782–2.353	0.281	0.278
Cake/donuts intake (daily intake = ref)	1.00			
Non-daily intake	0.757	0.452–1.267	0.263	0.290
Chocolates/candy intake (daily intake = ref)	1.00			
Non-daily intake	0.882	0.584–1.334	0.211	0.553

***** Based on activity energy expenditure of below or above 1680 METs-min/week. Low active was used as a reference category. aOR = adjusted odds ratio; CI = confidence interval; ref = reference category; SEE = standard error.

## Data Availability

All data generated or analyzed during this study are included in this published article. Any additional data will be available from the corresponding author upon reasonable request.

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
