# Peer review of "Anthropometric Measurements, Sociodemographics, and Lifestyle Behaviors among Saudi Adolescents Living in Riyadh Relative to Sex and Activity Energy Expenditure: Findings from the Arab Teens Lifestyle Study 2 (ATLS-2)"

_nutrients, 2021, doi:10.3390/nu14010110_

Round 1
Reviewer 1 Report
No comments
Really is a good paper
Author Response
Responses to Reviewer 1:
No comments from reviewer-1.
We thank the reviewer for considering to review our manuscript.
Reviewer 2 Report
Dear Authors, Thank you for submitting the manuscript: Anthropometric measurements, socio-demographics, and lifestyle behaviors among Saudi adolescents living in Riyadh relative to gender and activity energy expenditure: Findings from the Arab Teens Lifestyle Study 2 (ATLS-2). I hope you find the following commentary constructive: Abstract: Please add information on participants' age range. Introduction: The study is conducted in Riyadh. Can the selected area be considered typical, representative for Saudi Arabia? What factors decided about its choice as the place to conduct the study? General remark: In my opinion, the title and Introduction suggest that the research will include a group of adolescents. While it was only in the Results that it turned out that the study covered a group of 16-year-olds. The authors could specify it more in the methodology or even in the title of the article. Results: Some of the data are broken down by gender, and some by the level of physical activity. It is a bit incomprehensible to me. Table 6: I do not understand the first analysis due to age. It seems that all participants were of similar age? The same for Table 7: What was the "younger age" - no information on this in methodology section? General remark: In Discussion/Conclusions Authors could he authors coud propose some suggestions to promote physical activity and healthy diet in community. References: Format not always compliant with the requirementsAuthor Response
Reviewer’s Comments:
Abstract:
Please add information on participants' age range.
Authors’ Response:
We added the mean and SD of the age in the abstract section and age range in the result section.
Reviewer’s Comments:
Introduction:
The study is conducted in Riyadh. Can the selected area be considered typical, representative for
Saudi Arabia?
3
Authors’ Response:
Yes, the city is the capital of Saudi Arabia with over 6 million inhabitants and composed of people
from all over the country.
Reviewer’s Comments:
What factors decided about its choice as the place to conduct the study?
Authors’ Response:
As we said in a previous point, it is the largest city in the country and for logistic reason as we live in
it, we can conduct the study and supervise the data collection.
Reviewer’s Comments:
General remark:
In my opinion, the title and Introduction suggest that the research will include a group of
adolescents. While it was only in the Results that it turned out that the study covered a group of 16-
year-olds. The authors could specify it more in the methodology or even in the title of the article.
Authors’ Response:
The title indicated that the study involved adolescents and in the abstract awe added the mean age
and in the results we added the age range as well as the mean and SD of age.
Reviewer’s Comments:
Results:
Some of the data are broken down by gender, and some by the level of physical activity. It is a bit
incomprehensible to me.
Authors’Response:
Yes, we analyzed the data, as the title suggests, according to sex (males versus females) and activity
levels (low versus high active)
Reviewer’s Comments:
Table 6: I do not understand the first analysis due to age. It seems that all participants were of
similar age?
Authors’ Response:
No, the adolescent were from 14 to 19 years of age.
Reviewer’s Comments:
The same for Table 7:
What was the "younger age" - no information on this in methodology section?
Authors’ Response:
4
Please see our answer to the previous question. The analysis in indicated that we adjusted for age,
sex and SES in the logistic regression analysis.
Reviewer’s Comments:
General remark: In Discussion/Conclusions
Authors could he authors coud propose some suggestions to promote physical activity and healthy
diet in community.
Authors’ Response:
It seems that these suggestion are beyond the scope of the present study. Beside it will make the
discussion/conclusion very long. The other reviewer suggested reducing the discussion content.
Reviewer’s Comments:
References: Format not always compliant with the requirements
Authors’ Response:
We checked the reference format and it was fine
Reviewer 3 Report
The authors have investigated anthropometric measurements, socio-demographics, and lifestyle behaviours and their associations with physical activity levels in Saudi Arabian adolescents. Although the topic is potentially important and interesting, there are several methodological severe concerns, which should be sufficiently addressed to improve the quality of the manuscript.
1) The use of an unvalidated assessment questionnaire for physical activity, namely the most important variable, is the most serious methodological concern in this study. Each novel tool should be examined the validity and reproducibility before using it. Consequently, no one knows what is, in fact, measured while using this questionnaire. The authors should present the development procedure (e.g., refereeing to IPAQ) and the results of validation studies (e.g., comparison to the results derived from an accelerometer).
2) Abstract
Please provide information on the study period (as well as the methods section), age of participants, % of males, and the definition of overweight/obesity. The authors should explain how they estimated physical activity levels. According to the following reference, the use of "sex" is more appropriate than "gender" in this study.
Reference:
Heidari S, Babor TF, De Castro P, Tort S, Curno M. Sex and Gender Equity in Research: rationale for the SAGER guidelines and recommended use. Res Integr Peer Rev. 2016;1:2.
Methods
3) Participants and selection procedures
Please provide the number of eligible participants (i.e., those who were asked to participate in this study) and the number and the reason of participants excluded from the analysis. The reviewer recommends that the authors restrict the present analysis to the participants without missing variables. Thus, the number of participants included in Tables 1-3 must be 1189, and a re-calculated response rate may be provided in this section. A flowchart of selection procedures may help readers' understanding.
4) Assessment of lifestyle behaviours
Please provide information (e.g., correlation coefficients) on the validity and reliability of this questionnaire.
5) Dietary habits.
Please provide information (e.g., correlation coefficients) on the validity and the reproducibility of this questionnaire. Or discuss the degree and direction of influence by a lack of validation study on the present findings in the limitation section.
6) Line 185
A t-test for independent samples or chi-squared test for the proportion were used to test the differences between males and females in continuous or categorical variables, respectively.....? (as well as footnotes of tables)
7) Results
The results are too long. The authors do not have to all results in Tables. The descriptions in lines 207-213 are irrelevant because the comparisons were conducted between sex, not school type.
8) Tables
All table titles should be self-explanatory; the authors add the study name and period as well as the number and age of participants (except for tables including these values). Tables 1-3 can be combined to one table while "PA in Metabolic equivalent" should be deleted because the variable was duplicated in Table 4. Please add appropriate units for "Body shape index" and "Overweight/obesity" to Table 1. For Table 4, physical activities (min/week) should be deleted because these variables are included in those of METS-min/week. Please re-order variables in each activity (e.g., walking) as readers can understand which activities are included in moderate/vigorous activities. The authors should combine Tables 5-7 and provide one table showing the results derived from the multivariable analysis. Repeating similar analyses introduce a type 1 error. Please confirm the font of numbers in tables.
9) Discussions
Although discussion should be conducted depending on the revised results, the present discussion is also too long. Please summarise it to half. Besides dietary habits, the authors should not only point out the existence of limitations but discuss the degree and direction of influence on the present findings.
Author Response
Responses to Reviewer 3:
Thanks to the reviewer for the comments, which will improve the manuscript content. We have adopted the majority of the reviewer’s comments and suggestions. However, we have different opinions for few suggestions and we expect that such opinions to be respected. All the changes made to the revised manuscript were highlighted by yellow mark or track changes.
Reviewer’s comments:
1) The use of an unvalidated assessment questionnaire for physical activity, namely the most important variable, is the most serious methodological concern in this study. Each novel tool should be examined the validity and reproducibility before using it. Consequently, no one knows what is, in fact, measured while using this questionnaire. The authors should present the development procedure (e.g., refereeing to IPAQ) and the results of validation studies (e.g., comparison to the results derived from an accelerometer).
Authors’ Response:
We used a validated and reliable assessment tool. We also included some references to the development, reliability, and validity of the ATLS questionnaire in previous studies (references 5, 26, 27, & 28). There is no need to replicate the results of the validation papers that have been already published in open access journals.
Reviewer’s comments:
2) Abstract
Please provide information on the study period (as well as the methods section), age of participants, % of males, and the definition of overweight/obesity. The authors should explain how they estimated physical activity levels. According to the following reference, the use of "sex" is more appropriate than "gender" in this study.
Reference:
Heidari S, Babor TF, De Castro P, Tort S, Curno M. Sex and Gender Equity in Research: rationale for the SAGER guidelines and recommended use. Res Integr Peer Rev. 2016; 1:2.
Authors’ Response:
We are limited to 200 words in the abstract section. Therefore, giving a lot of methodological details is not possible in the abstract. However, we added the age and percentage of males in the abstract section. All other information (definition of overweight or obesity and estimated physical activity levels) were already provided in the methods section.
As to replacing gender with sex, it was replaced throughout the texts.
Reviewer’s comments:
Methods
3) Participants and selection procedures
Please provide the number of eligible participants (i.e., those who were asked to participate in this study) and the number and the reason of participants excluded from the analysis.
The reviewer recommends that the authors restrict the present analysis to the participants without missing variables. Thus, the number of participants included in Tables 1-3 must be 1189, and a re-calculated response rate may be provided in this section.
A flowchart of selection procedures may help readers' understanding.
Authors’ Response:
The response rate was extremely high, reaching 99.2%. We reported these information under the results section.
We added the following statement under the participants and selection procedures: “Within the selected classes, all healthy students without any medical condition that preclude them from physical activity were invited to participate in the study.”
As to the missing cases, we have a really low percentage of missing cases (5.7%) and they were randomly distributed among all physical activity choices and both sexes. We have also checked earlier the mean values of the descriptive characteristics (age, weight, height, BMI, waist circumference, waist to height ratio, body shape index, and overweight/obesity prevalence) and we found no significant differences between those reported to the whole group (N = 1260) and those reported in table 4 (N = 1189). In fact the figures are almost the same.
We added a flowchart for the selection procedures, as suggested by the reviewer (figure 1).
Reviewer’s comments:
4) Assessment of lifestyle behaviours
Please provide information (e.g., correlation coefficients) on the validity and reliability of this questionnaire.
Authors’ Response:
As we mentioned in point # 1, the psychometric properties of the instrument that we have used have been reported earlier in the references 5, 26, 27, & 28.
Reviewer’s comments:
5) Dietary habits.
Please provide information (e.g., correlation coefficients) on the validity and the reproducibility of this questionnaire. Or discuss the degree and direction of influence by a lack of validation study on the present findings in the limitation section.
Authors’ Response:
The validity and reliability of the dietary habits that were part of the questionnaire were reported earlier by the following reference, which we have added in the references list.
Musaiger AO, Bader Z, Al-Roomi K, D’Souza R. Dietary and lifestyle habits amongst adolescents in Bahrain. Food Nutr Res 2011; 55. DOI:10.3402/fnr.v55i0.7122.
Reviewer’s comments:
6) Line 185
A t-test for independent samples or chi-squared test for the proportion were used to test the differences between males and females in continuous or categorical variables, respectively.....? (as well as footnotes of tables)
Authors’ Response:
Thanks for this correction. We modified these sentences accordingly in the statistical analysis section as well as in table 1.
Reviewer’s comments:
7) Results
The results are too long. The authors do not have to all results in Tables. The descriptions in lines 207-213 are irrelevant because the comparisons were conducted between sex, not school type.
Authors’ Response:
We have deleted some words/phrases in the results section and reduced the words count accordingly.
The findings that reported for school type relative to activity levels (low versus high active) is in table 4 (in the revised MS) and represent 2 by 2 cross tabulation and we believe that it is relevant to reporting of the results.
Reviewer’s comments:
8) Tables
All table titles should be self-explanatory; the authors add the study name and period as well as the number and age of participants (except for tables including these values).
Tables 1-3 can be combined to one table while "PA in Metabolic equivalent" should be deleted because the variable was duplicated in Table 4.
Please add appropriate units for "Body shape index" and "Overweight/obesity" to Table 1.
For Table 4, physical activities (min/week) should be deleted because these variables are included in those of METS-min/week.
Please re-order variables in each activity (e.g., walking) as readers can understand which activities are included in moderate/vigorous activities. The authors should combine
Tables 5-7 and provide one table showing the results derived from the multivariable analysis. Repeating similar analyses introduce a type 1 error.
Please confirm the font of numbers in tables.
Authors’ Response:
All tables are self-explanatory.
We combined table 1 and 2 in one table (table 1 now). However, table 3 (now table 2) has different style as it includes a heading entitled “Criterion”. Besides this will make table 1 very long table that is difficult to manage in editing process .
As to physical activity values in table 3 (in the current version and was table 4 in the previous version), they are reported differently as means and standard errors (in table 3), while those in table 2 (in the current version and was table 3 in the previous version) are reported as proportions above or below certain cut-off values.
We added the units for overweight or obesity in table 1, but the body shape index has no unit, as it is a combination of waist circumference (cm), BMI (raised to 2/3), and height raised to ½.
In table 3 (4 in the previous version), we deleted physical activity in min/week.
We added a legends to table 3 showing what types of activities are included in moderate-intensity or vigorous-intensity activities.
It is true that table 5 and 6 are both multivariable analysis procedures, however, each one provides specific and important analysis. For example table 5 presents interaction effects in selected variables between activity levels and sex and a couple of them were significant.
Reviewer’s comments:
9) Discussions
Although discussion should be conducted depending on the revised results, the present discussion is also too long. Please summarise it to half. Besides dietary habits, the authors should not only point out the existence of limitations but discuss the degree and direction of influence on the present findings.
Authors’ Response:
We have slightly reduced the word counts in the discussion section, however, you have to understand that we have a lot of findings to discuss, as the current study is not a short paper (6 full tables and many lifestyle variables were included).
If we are going to discuss the strength and limitations in depth, this will make the paper even much longer. Also, the limitations were stated and they are fairly straight forward.
Round 2
Reviewer 2 Report
Dear Authors,
Thank you to the author for revising the manuscript.